# Non fire-adapted dry forest of Northwestern Madagascar: Escalating and devastating trends revealed by Landsat timeseries and GEDI lidar data

Joseph Emile Honour Percival[1]*, Hiroki Sato[2], Tojotanjona Patrick Razanaparany[3], Ando Harilalao Rakotomamonjy[1,4], Zo Lalaina Razafiarison[3], Kaoru Kitajima[1,5]

1 Graduate School of Agriculture, Kyoto University, Kyoto, Japan, 2 Graduate School of Asian and African Area Studies, Kyoto University, Kyoto, Japan, 3 Faculty of Science, University of Antananarivo, Antananarivo, Madagascar, 4 Ecole Doctorale et Écosystème Naturel, University of Mahjanga, Mahjanga, Madagascar, 5 Smithsonian Tropical Research Institute, Balboa, Panama

* ipercival@gmail.com

**Data Availability Statement:** The minimal data set associated with this study available at: 10.6084/ m9.figshare.24922098.

## Abstract

Ankarafantsika National Park (ANP), the last significant remnant of Northwestern Madagascar's tropical dry forests, is facing rapid degradation due to increased incidences of fire. This poses severe threats to biodiversity, local livelihoods, and vital ecosystem services. Our study, conducted on 3,052-ha of ANP's pristine forests, employed advanced remote-sensing techniques to assess fire impacts during the past 37 years. Our aims were to understand historical fire patterns and evaluate forest recovery and susceptibility to repeated fires following initial burns. Using data from multiple Landsat satellite sensors, we constructed a time series of fire events since 1985, which revealed no fire activity before 2014. The Global Ecosystem Dynamics Investigation (GEDI) lidar sensor data were used to observe forest structure in both post-fire areas and undisturbed zones for comparison. We recorded six fire incidents from 2014–2021, during which the fire-affected area exponentially grew. A significant fire incident in October 2021 impacted 1,052 hectares, 59% of which had experienced at least one fire in two-to-four years prior, with 60% experiencing two preceding incidents: one in 2017 and another in 2019. The initial fire drastically reduced plant cover and tree height, with subsequent fires causing minor additional loss. Post-fire recovery was negligible within the initial four years, even in patches without recurrent fires. The likelihood for an initial burn to trigger subsequent fires within a few years was high, leading to larger, more severe fires. We conclude that ANP's dry forests exhibit high vulnerability and low resilience to anthropogenic fires. Prompt preventive measures are essential to halt further fire spread and conserve the park's unique and invaluable biodiversity.

## Introduction

Tropical dry forests, which generally occur in areas with mean annual temperatures exceeding 17°C with both prolonged dry season and wet season [1], are among the most fire-prone

**Funding:** The authors KK and HS received funding from the Japan Society for the Promotion of Science for the following grants: "Fostering Joint International Research (B)" with grant number 18KK0179, "Grant-in-Aid for Scientific Research (A)" with grant number 22H00424, and "Grant-in-Aid for Scientific Research (B)" with grant number 22H03837. The Japan Society for the Promotion of Science's website can be accessed at https://www.jsps.go.jp/english/e-grants/. KK and HS were also awarded the grant "JST aXis" with grant number JPMJAS2013, funded by the Japan Science and Technology Agency, whose website is https://www.jst.go.jp/global/axis/en/index.html. HS and KK received support from the Heisei Memorial Research Grant, funded by The Japan Prize Foundation, with information available at https://www.japanprize.jp/en/foundation_activities.html. Additionally, funding support for the project was provided by the Kyoto University SPIRITS Fund, which was facilitated by Yusuke Onoda. Information about this fund can be found at https://www.kura.kyoto-u.ac.jp/en/support/ekkyo/spirits/. The funders had no role in study design, data collection and analysis, decision to publish, or preparation of the manuscript.

**Competing interests:** The authors have declared that no competing interests exist.

biomes on Earth. However, natural fire regimes exhibit wide variations within this biome, which pose significant challenges to the maintenance of ecological integrity, ecosystem services, and human well-being. The increasing frequency, severity, and extent of fires in these ecoregions, often driven by a combination of natural and human factors such as droughts, high temperatures, land-use practices, and population growth, have led to extensive losses of forest cover, biodiversity, and soil fertility [2]. Fires can alter the ecological processes and functions of the ecosystems within these biomes, including nutrient cycling, water availability, and carbon sequestration, with profound implications for climate change mitigation and adaptation [3]. Furthermore, fires can have severe social and economic impacts on local communities, who rely on these ecosystems for their livelihoods and cultural identity. Hence, understanding the ecological and socio-economic impacts of fires toward development of effective management strategies to prevent and mitigate their occurrence is a pressing global challenge [4].

Madagascar, with its unique biodiversity and endemism, has experienced a significant loss of forests during the last several decades. Particularly the dry forests in NW Madagascar have seen a decrease in patch sizes and distance to non-forest edges more than any other forest types [5, 6]. Human-induced fires are the greatest threat to the remaining dry forests of Madagascar, which are highly valued for biodiversity conservation, local communities, and research [7]. The western part of the country, where the dry season often lasts over six months (Fig 1A) and people burn forests to expand agricultural and pastoral lands, has seen an average of over 350,000 fires annually between 2012 and 2019 [8]. As a result, nearly 50% of the remaining dry forests occur within 1-km of fire each year [6, 8], indicating the high likelihood of additional fires and a steady increase in the frequency and size of fires in recent years. These fires acutely threaten the fragmented landscape of dry forests in NW Madagascar, highlighting the urgent need for effective wildfire management strategies to protect these valuable ecosystems [7–9].

Although fire is a widespread phenomenon, its specific impact on Madagascar's dry forests, especially regarding initial fire events and subsequent recovery versus the risk of repeated burns, is yet to be quantitatively evaluated across suitable spatiotemporal scales. On-the-ground observations of dense natural forests in the Ankarafantsika National Park by the authors suggest that once burned, the same area becomes more vulnerable to subsequent burns. This is quite unlike many fire-adapted savannas and open-canopy dry forests, where repeated fires can lead to a reduction of ground litter, thus decreasing the likelihood of more severe fires [10]. Instead, first-time burns in non-fire adapted forests, or even in fire-adapted forests where fire has been suppressed over extended periods, can result in large amounts of fuel from partially killed trees and dead logs that can lead to high tree mortality rates and increased likelihood of future fires [11]. Several studies have suggested that repeated fires may significantly alter the forest structure, potentially leading to a transformation from dry forest to grassland and savanna in Madagascar's dry forests [9, 12, 13]. But this perspective is not unchallenged; fire may not be a consistent proxy for forest degradation [14], and some report that the savanna/dry forest landscape in Madagascar may be relatively resilient to fire and other disturbances [12, 15, 16].

The lack of comprehensive studies on the historical fire regimes of NW Madagascar leaves much to be inferred about fire's role in shaping these environments. However, evidence such as the presence of fire-adapted plant communities [13, 17] and historical charcoal deposits in soils [18, 19] indicates that fire is a longstanding component in many regions of Madagascar. For instance, a dramatic increase in charcoal in lake sediment in the coastal area of SW Madagascar around 2,000 BP, suggests early human impacts on vegetation and biodiversity [20]. Fire plays a significant role in shaping the landscape, but it remains uncertain how these impacts vary across different regions and ecosystems. Variations in fire regimes among forest

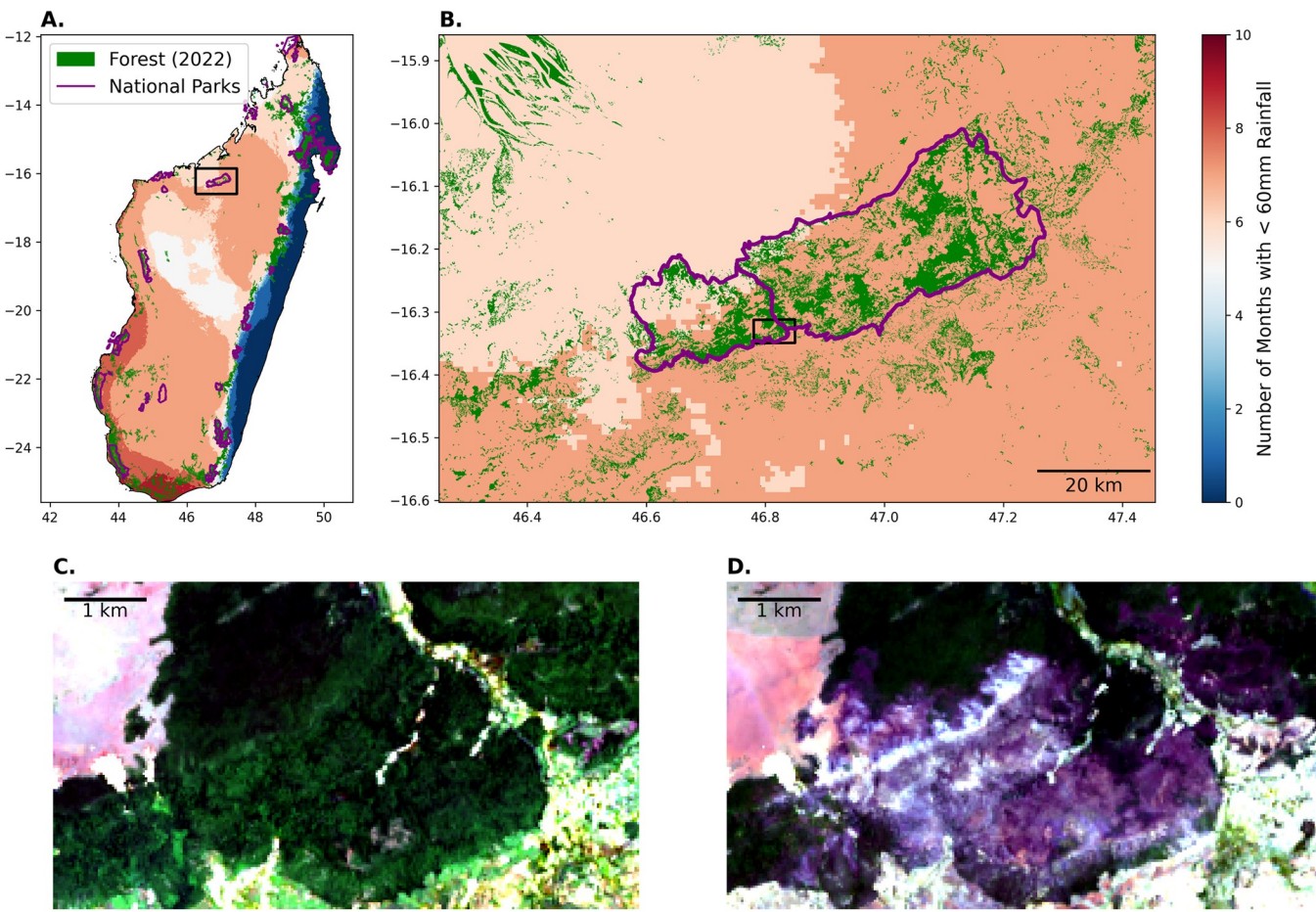

**Fig 1. Study area in Northwestern Madagascar.** Maps and satellite imagery showing: (A) Precipitation patterns in Madagascar as defined by the number of months with rainfall less than 60mm (precipitation data is from [33]) and forested area as of 2022 (following the method of [5]); (B) enlarged panel of Ankarafantsika National Park; and (C) the study area on January 26, 2014 and (D) on December 15, 2021. Images in (C) and (D) are Landsat-8 images courtesy of the U.S. Geological Survey.

types, each of which with unique assembly of plants and animals, may exist side-by-side in NW Madagascar.

Fire regimes may display drastic differences even within a few kilometers across the landscape. For example, in sand-dominated landscapes of Florida, mature vegetation shifts from longleaf pine savannas (sand hills), maintained by frequent summer fires every two-to-five years to various types of pine- or oak-dominated forests that differ in fire frequency and intensity [21, 22]. Moreover, the geographical expansion of humans has been linked to shifts in fire regimes over archaeological timescales spanning thousands to tens of thousands of years [23]. In many parts of the world, tropical and subtropical savannas are dominated by fire-tolerant trees that coexist with grass and other herbaceous plants. These "old-growth savannas" require frequent fire for their maintenance [24]. Fire suppression, on the other hand, may encourage the invasion of fire-intolerant trees, leading to changes in vegetation and ecosystem properties [25]. Conversely, tropical forests historically devoid of fire may be vulnerable to human-induced changes in fire regimes, potentially resulting in biodiversity loss and reduced carbon storage capabilities [26, 27]. To ensure the optimal conservation of Madagascar's biological and cultural diversity, it is urgent to conduct ecological studies that identify differences in

types of dry forests and savannas with contrasting historical fire regimes necessary for their sustainability. Further comprehensive and quantitative assessment of fire impacts in tropical dry forests in Madagascar is needed, entailing detailed examination of fire histories, seasonality, and tree communities at the forest stand levels of tens to hundreds of hectares.

Here, we report an analysis of fire impacts on a tropical dry forest within Ankarafantsika National Park (ANP) (Fig 1B). Established 1927, it is one of the first five national parks in Madagascar and holds the largest remaining fragment of the dry forest ecosystem in western Madagascar [28]. Historically, areas surrounding ANP also had substantial areas of dry forests, but their coverage has declined rapidly, especially during the last few decades, due to fires, many of which were set on purpose and often spread into the protected zones within the park [29–32]. Prompted by a major fire episode of 2021 that we witnessed on the ground, which was set off as arson near the southern border of the park [31], we set out to evaluate the recent history of fire in ca. 3,000-ha. This area used to hold a contiguous semi-deciduous forest with a closed canopy until several years ago (on-the ground observation by Hiroki Sato, Fig 1C and 1D). This dry semi-deciduous forest in the area near Ampijoroa Forest Station and Ambodimanga village not only has high values for biodiversity conservation as a habitat for eight species of lemur and several endemic reptiles and birds, but also contribute to local livelihood, ecotourism, and scientific research.

The main goals of the study are to (1) understand the historical patterns of fire in the region, and (2) evaluate the degrees of recovery and vulnerability to repeated fires within several years following the first. Fire history was mapped with Landsat timeseries since 1985 and ultra-high resolution Planet Scope imagery [34]. Forest structural attributes included percentage of canopy cover, total plant area index, and canopy height as measured by the Global Ecosystem Dynamics Investigation (GEDI) space-based lidar sensor [35]. We tested the association between fire history and forest structure using linear mixed models and characterized differences between areas with different burn histories. In the 3,052-ha area examined, no fire was observed before 2014 and the vast majority of burning took place in 2017, 2019, and 2021. Hence, the dataset included areas with zero, one, two, and three burns, allowing us to examine the relationship between the number of fires and the time since the last fire on forest structure attributes.

## Results

### Fire records from 1985 to 2022

Our study reveals that fires emerged as a recent phenomenon within the 3,052-ha protected forest area near Ampijoroa Forest Station and Ambodimanga village, with the first observation of minor fires in 2014. Over the period of 37 years from 1985 to 2022, we detected a total of six fires (Fig 2B). The two fire occurrences in 2014 were relatively minor, affecting approximately 40-ha, but subsequent fires in 2017, 2018, 2019, and 2021 burned rapidly increasing areas, with the largest and most recent fire having burnt over 1,034-ha (Fig 2B). Four of the six fires during the last five years were responsible for approximately 98% of the total area burned. All fires took place during the late dry season, with most of the burned area located within the park's buffer zone.

The fire history summarized in Fig 2 indicated the occurrence of up to zero, one, two, or three distinct fire events in any given location within our study site (Fig 2A). We found that 23% of the original forest area considered by our study (2,152-ha) experienced a single fire episode, 12% experienced two, 18% experienced three, whereas 47% experienced no fires at all. Additionally, we found that there was a significantly higher likelihood of an area burning again after its first fire; nearly 60% of the burnt area had been burnt one-to-five years prior.

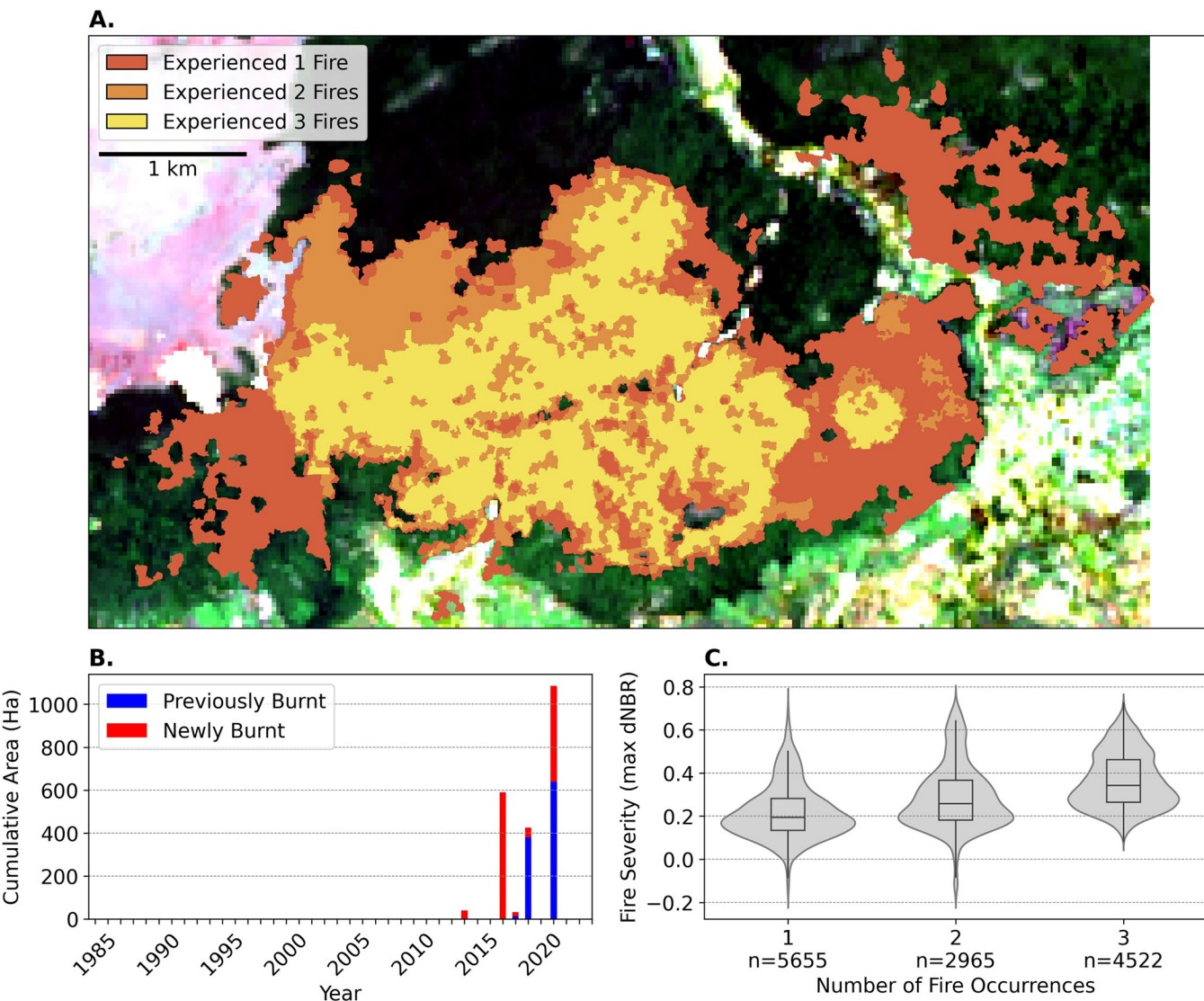

**Fig 2. Fire history within the 3052-ha study area of Ankarafantsika National Park.** (A) The number of times fire has occurred (one, two, and three times indicated by color; Landsat-8 images courtesy of the U.S. Geological Survey); (B) the total area burnt for each type of burn history for each year; (C) burn severity as measured by the maximum detected differenced normalized burn ratio (max dNBR) in relation to the total number of fires experienced with 'n' being the number of pixels observed for each burn count.

Finally, burn severity, evaluated by the differenced Normalized Burn Ratio (dNBR), which measures changes in vegetation reflectance indicative of fire impact, revealed that burn severity generally increased with each successive burn (Fig 2C).

### Fire impacts on forest structure

There were significant differences in plant area index (Kruskal-Wallis test, H = 310.67; $p < 0.001$), canopy cover (H = 310.75; $p < 0.001$), and canopy height (H = 129.92; $p < 0.001$) between areas with different numbers of fire occurrences (Fig 3A–3C). The Dunn tests further showed that these differences were mostly between areas that had had no fire versus those that had experienced at least one fire ($p < 0.001$). Repeated fires caused little further deteriorations in forest structure detectable from the remotely sensed data, with no significant differences in

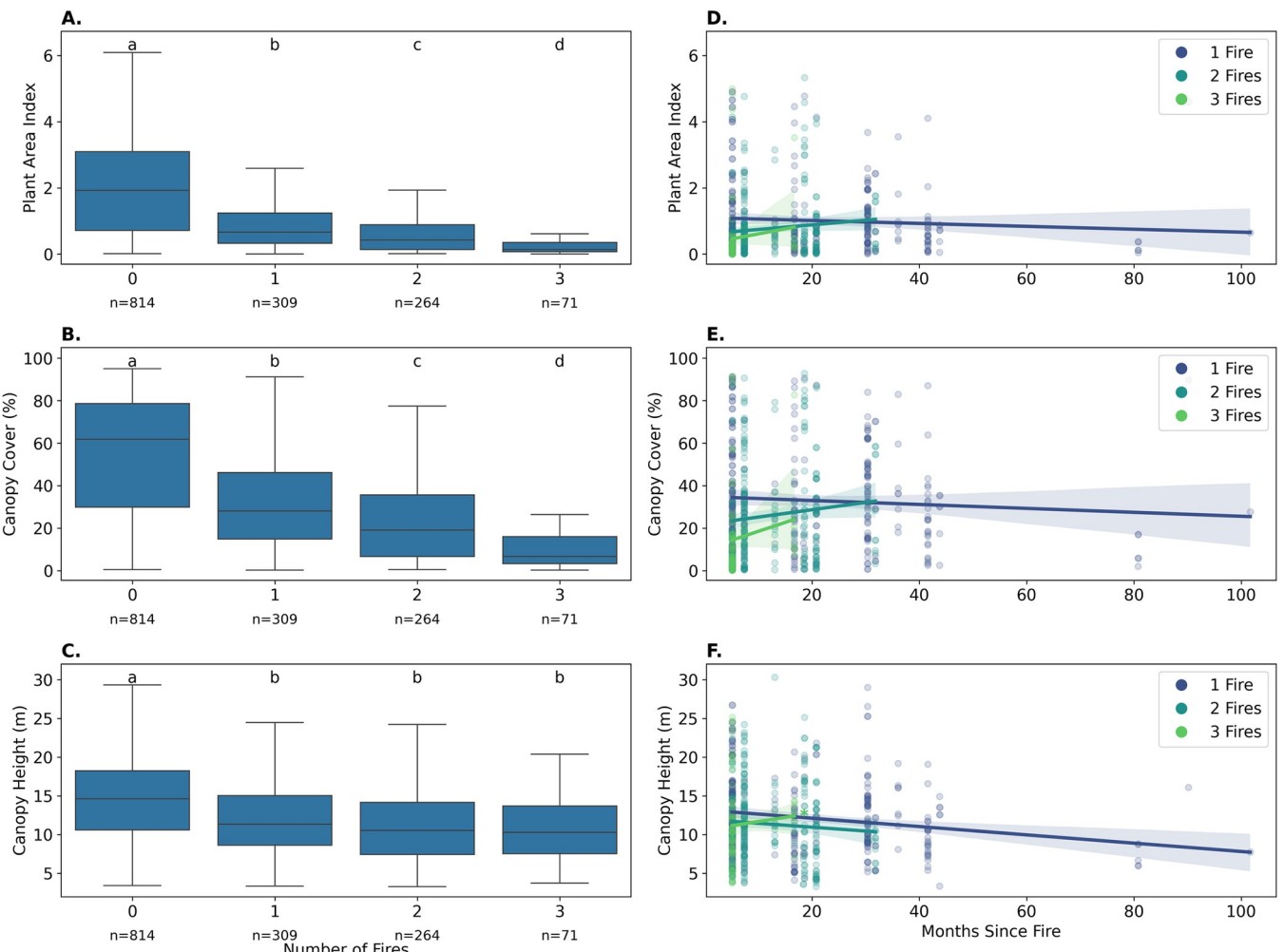

**Fig 3. Differences in forest structure across varying burn histories.** Differences in forest structure as described by plant area index (A), canopy cover (B), and canopy height (C), across the number of fire occurrences; and over months since the last fire (D-F), with 'n' denoting the number of data points from the Global Ecosystem Dynamics Investigation (GEDI) lidar sensor observed per factor. Groups in subplots A, B, and C not sharing lowercase letters are significantly different each other (p < 0.05) based on Dunn's post hoc tests.

canopy height between areas with one fire versus two or more fires (Fig 3A–3C, S1 Table). Significant further decrease of plant area index and canopy cover was detected between one, two, and three fires.

In examining the impact of time since the most recent fire (hereafter, time since fire), our linear mixed models revealed that there was no evidence of notable recovery within the first four years after a fire. Concurrently, the frequency of fires exhibited a pronounced and consistently negative on forest structure (Fig 3D–3F, S2 Table). Time, denoted by number of months, since fire had no significant effect on either plant area index or canopy cover. A weak but statistically significant negative effect of time since fire on canopy height (coefficient = -0.041, p = 0.02) reflects the fact that trees left standing during the first fire fall over time. As indicated by the pooled data regardless of the time since fire (Fig 3A–3C), the model outputs (S2 Table) lender statistical support of the negative effects of repeated fire on plant area index (coefficient = -0.2, p = 0.004), canopy cover (coefficient = -0.071, p < 0.001), and canopy height (coefficient = -0.114, p = 0.013). The random effects of precipitation seasonality and park

management zone in these models showed little variance, thus indicating that their inclusion in the models hardly affected the model outcomes (S2 Table).

## Discussion

The escalating frequency and size of dry season fires poses a serious threat to plant and animal habitats in our study area, eliciting deep concern among the managers of Ankarafantsika National Park. This issue also reverberates with the local communities, especially those in Ambodimanga village located within the park, who are legally allowed to use community forest zone within the park and rely on its ecosystem services for their livelihood and well-being [28, 36]. Our study is the first to show quantitatively that extensive and repeated forest fire in this area is a recent phenomenon in the last five years. Furthermore, the results show the first burn imparts serious damage to the ecosystem and leads to likely fire returns in just few years.

Informal interviews of national park staff and residents conducted by some of us indicated that forest fires are often instigated by individuals attempting to establish agricultural and grazing lands, or as acts of protests to park regulations or government policies. Population growth, including the influx of migrants mainly from Southwest Madagascar, is driving the increased demand for food and energy resources in both rural and urban areas. While some of these fires may have been started outside the park border, some spread into the park. Reports from other parts of western Madagascar indicate that, since the 2000s, fire incidents have become more frequent and widespread [37, 38]. Such anthropogenic fires are likely to differ from fires naturally ignited by lightning strikes. In the study region, lightning strikes are considerably more frequent during the wet season [12, 19], which is the season when the forests are too wet to burn. Hence, we argue that forest fires caused by lightning strikes were historically rare in the study site, and disruptive anthropogenic fires in the dry season is a novel threat to non-fire adapted forest ecosystem of the region in association with population growth. Changing climate, with rising temperatures, increased drought, and stronger winds, may exacerbate fire frequency and alter the resilience of forests [39–42]. Therefore, it is important to thoroughly understand the impacts of fire on the park's biodiversity and ecosystem services.

Our findings indicate that there is very limited forest regrowth within the first four to five years following a fire (Fig 3D–3F). Studies on forest fire recovery in fire adapted vegetation in both dry tropic and temperate zones typically suggest that leaf area index (which is closely related to plant area index) recovers faster than other attributes of forest structure and may return to pre-fire levels within two to nine years–with forests in fire-prone areas typically displaying much higher resilience and more rapid recovery [43–45]. Indeed, many fire adapted dry forests require fire regimes that cycle every few years [21, 22]. However, we observed little-to-no regrowth after four or more years (our data went up to over 8 years) post-fire. This seems contrary to the results of a study conducted in Central Menabe, 580-km south of our study site, where complete forest recovery within twelve years of a fire was observed [15]. This discrepancy may be due to differences in plant community composition (i.e., the proportion of fire tolerant species; as is discussed in [13]), or to the differences in abiotic factors such as soil compositions in each location. From our analyses we suggest that the forests in Ankarafantsika largely consist of trees with no fire tolerance and represent an "old-growth dry forest" with closed canopy that is quickly transformed by a single fire to "degraded savannas/grasslands" with low biological diversity [24]. Whereas the study area of [15] consisted of dark and yellow soils, our study was conducted in an area consisting mainly of quartz-rich white sand (i.e., derived from Jurassic coastal sand dunes), classified as Quartzipsamment [46], which severely restrict water and nutrient availability (see also [47]). Such infertile soils select for nutrient-conservation strategies, including evergreen leaf habits, in the absence of fire, rather than fire-

adaptive strategy of dry season deciduousness and thick bark, which explain the low resilience of these forests to fire.

Visual inspections and vegetation survey on the ground help explain the scant recovery of vegetation reported in our study (Fig 3). Lack of species with fire adaptive traits, together with the disappearance of dense roots in the upper soil zone following a fire (unpublished data of Kaoru Kitajima and colleagues), could potentially instigate an ecological transformation of forest dominated by late-successional evergreen broadleaf trees to secondary vegetation dominated by pioneer vines and shrubs, and ultimately to savanna with grass and bare grounds. Even after a few repeated fires, our results indicate a potential that savanna grasses will gain a competitive edge over the evergreen trees that, while not strictly dominant, comprise a significant proportion of these forests when undisturbed. After a single fire event, the altered conditions seem unfavorable for the recovery and resurgence of these evergreen species, prompting a major shift of the community composition. Consequently, it is essential to conduct further research to assess the influence of both soil characteristics and changes in plant community composition on the post-fire recovery process.

The findings of this study also demonstrate that relatively small fire incidents burning less than tens of hectors may potentially have devastating impacts on the dry forest ecosystem in Madagascar, initiating vegetation shift. Such fires can be a catalyst for larger, more severe fires, because burning of previously unburned forest is frequently followed by repeated fires just a few years later, which tend to be more destructive. These fires sweep through the forest, leaving behind both standing and fallen dead trees (Fig 4). These remnants then serve as kindling for future fires (Fig 4B and 4C), establishing a positive feedback loop that fuels further fires [11]. While we cannot definitively conclude that a single fire is sufficient to alter the ecosystem, it alters the fire regime irreversibly. This is consistent with the reporting by [6], who suggested that a second fire within three years of the first can result in a more intense fire and open canopy, facilitating the success of grasses, such that three to four fires within a 15-year period may transform a forest into grassland. Our analysis also revealed that canopy cover and plant area index were more affected than canopy height after one or more fires, indicating that the fire opens the canopy while leaving some trees standing. Ground-based observation of the site one to two years after the major fire of 2021 suggests that most tall trees do not appear to produce canopy leaves again after the first burn (Fig 4). This pattern is visible in drone photo survey repeated by us as well. Instead of recovering, these remaining tree stems, appearing mostly dead, provide the high fuel load needed for further fires and pave the way for ecosystem shifts.

Looking forward, we identify two key priorities: 1) finding ways to prevent further fires within the national park and 2) investigating the effects of fire on the biological community composition within the park. Prescribed fires have been tried outside the park as a management tool to create fire breaks to prevent fires from spreading from savanna into forests [8, 12, 14]. But they have limited effectiveness in preventing fires that originate within the park with high fuel load. Further research is needed to determine the most effective methods for mitigating future fires, such as the optimal design of setting fire breaks in the buffer zone or the role of wider trails that can double as a fire break and an ecotourism infrastructure (see [48] and Fig 4A). Madagascar is home to some of the most unique ecosystems with extremely high levels of biodiversity and endemism, with many understudied species specializing in dry deciduous forests, including reptiles, insects, small mammals, and aquatic organisms [49]. Clearly, there is an urgent need to investigate how fires impact populations of these species. Dry forests inside the Ankarafantsika National Park, provide important ecosystem services by preventing soil erosion, protecting belowground aquifers, and provisioning natural resources [50]. It is critical that additional studies are conducted across the entire National Park to understand the potential increasing threat of fire on the entire park. Given both the local and global

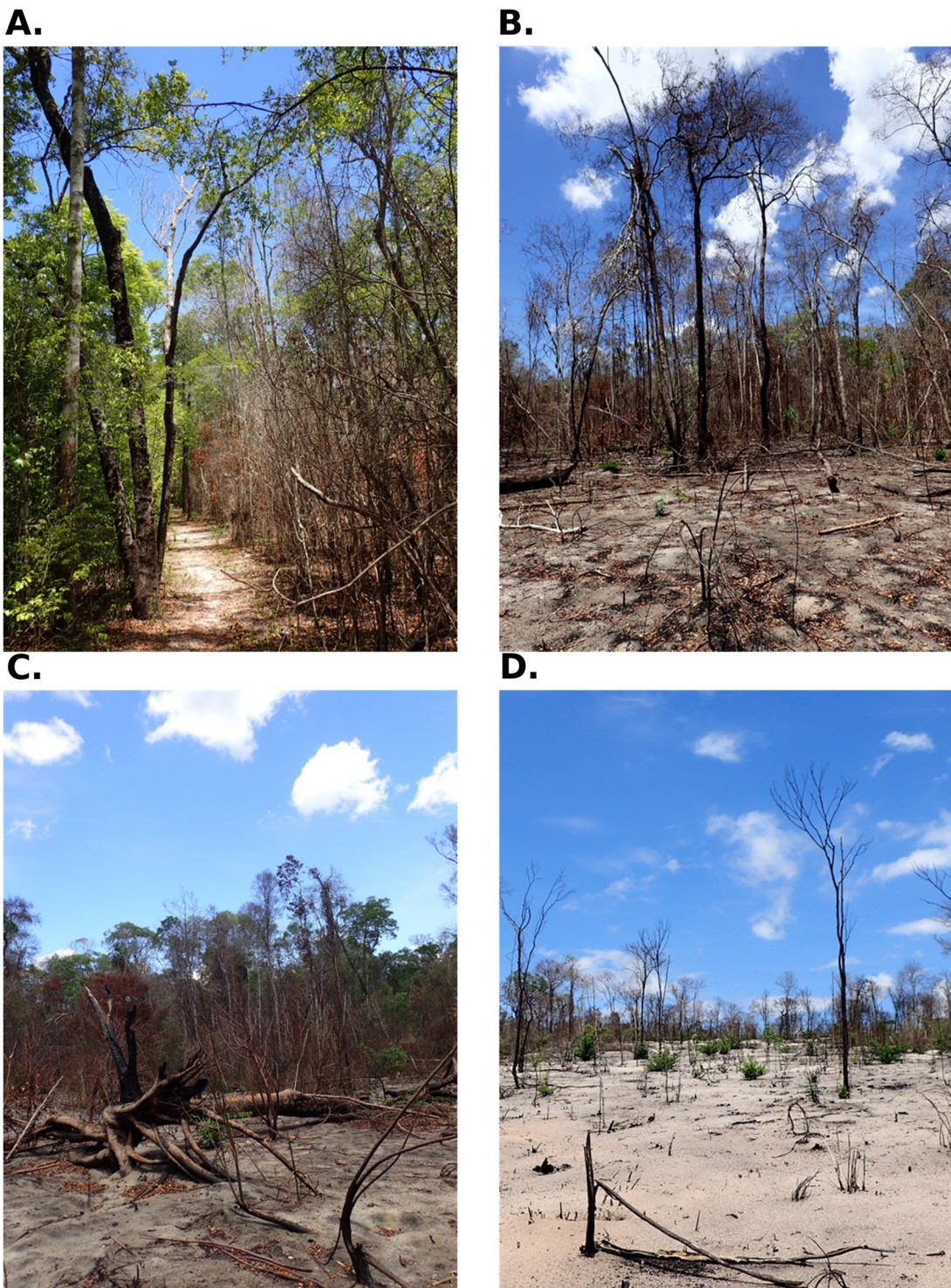

**Fig 4. Typical conditions of vegetation in the study area after the fire in October 2021.** (A) Non-burnt and once burnt area at left and right of an ecotourism trail that served as a firebreak, (B) Boundary of twice burnt area (front) and once burnt area (back), (C) Unburned dead wood that will become fuel for the next fire, (D) Third burnt area with bare white sand.

importance of dry forests in Madagascar, understanding their resilience to fire and fire interactions is essential in setting priorities to protect these forests.

## Methods

### Study site

The present study was conducted in the vicinity of Ampijoroa Forest Station and Ambodimanga village, within Ankarafantsika National Park. This region has held protective status since 1927, initially as the Ankarafantsika Natural Reserve, before being officially recognized as a national park in 2001 [28, 50]. The park spans an area of approximately 136,513 hectares and stands as the largest intact fragment of the western dry forest ecosystem in Madagascar [28]. Local villagers inside the park, who had agreed to limit the expansion of their cropping areas when the protected area was established in 1927, conduct paddy rice production in the valleys [16].

The park's forest is characterized by a diverse mix of deciduous and evergreen broadleaf trees. Satellite photos indicate that type of vegetation was also common outside of the park boundary until 1980's and 1990's. However, its boundary is now delineated by savannas and grasslands, that are frequently subjected to slash-and-burn cropping, charcoal production, and zebu cattle grazing.

The region experiences a prolonged dry season from May to October, during which rainfall is typically less than 10 mm [51]. In contrast, the wet season, running from November to April, sees an average rainfall of around 1,400 mm [51]. Although fire incidents are common in the areas surrounding the entire park, the fire regime inside the park including the study area is described as low and variable according to [14].

The park is systematically divided into five zones—peripheral, community, buffer, service, and core—each with distinct regulations and uses [16]. The service zone is dedicated to biological and ecological research and ecotourism. For this study, we focused on a total of 3,052 hectares spanning the service zone, the buffer zone, and the community zone. The northern portion of the study area included the service zone that encompasses the Ampijoroa Forest Station, a hub for ecotourism and academic research activities. Conversely, the southern region, around Ambodimanga Village, is marked by a buffer zone extending to the west, and a sustainable 'community' use zone to the east. In this 'community' use zone, villagers are permitted to engage in subsistence activities such as gathering forest resources and small-scale farming [28].

### Data

This study employs remotely sensed data from diverse sources and leverages a variety of open-source software and libraries, chiefly in R and Python. We generated fire history maps using monthly median spectral composites of Landsat 5, Landsat 7, and Landsat 8 data, as well as 4-band very high-resolution Planet imagery [34], and aerial imagery captured by a small aircraft (in 1990) and a drone (in 2021). We conducted a statistical analysis of the impact of fire history on forest structure utilizing forest structural attributes sourced from the Global Ecosystem Dynamics Investigation (GEDI) [35], a space-based lidar sensor hosted on the International Space Station that captures three-dimensional biophysical attributes of forest structure within 25-m diameter footprints.

### Fire history maps

We produced fire history maps through a three-phase process. Initially, we collected data and established time series for each Landsat pixel (30-m by 30-m). Subsequently, we employed an

unsupervised, rule-based classification system to map probable instances and locations of fires. Finally, we acquired higher-resolution data to generate supervised fire-non-fire classified maps at the potential times of fire.

In the initial phase of the study, we analyzed Landsat data in Google Earth Engine to produce monthly median pixel composites of the Normalized Difference Vegetation Index (NDVI) and the Differenced Normalized Burn Ratio (dNBR) since the year 1985. These composites were then exported for time series analysis. A binary map of forest cover for the year 1990 was constructed utilizing a supervised object-based classification algorithm, which incorporated segmentation using the Simple Linear Iterative Clustering algorithm [52] and a Multi-Layer Perceptron artificial neural network from the Scikit-Learn python library [53] for classification. We used this as an initial forest base map to "mask" the NDVI and dNBR time series data, a process which involved filtering the data to only include areas identified as forest cover, resulting in over 60,000 time-series objects for both NDVI and dNBR. We chose the year 1990 as a base map due to the availability of high-resolution aerial photographs of the region from this year. Visual comparison between satellite imagery from 1985 and 1990 yielded little differences in forest cover.

In the second phase, we de-trended the NDVI and dNBR time series data using the Breaks For Additive Season and Trend (BFAST) algorithm [54, 55]. We also utilized the BFAST algorithm to detect breaks in trend and wrote an R program to match negative breaks in trend with spikes in dNBR. Based on the seasonal trend of NDVI and dNBR in the region, we established a dNBR threshold value of 0.2 to set criteria for fire detection. These criteria stipulated that any negative break in NDVI trend, accompanied by a dNBR value of 0.2 or greater within the 95% confidence interval of the break, would be classified as a fire. To reduce noise caused by persistent clouds, shadows, and missing values, we grouped contiguous pixels into polygons and eliminated any unsupervised classified fire area less than five hectares.

In the final phase, we queried the Planet database for images taken during each probable fire period, and manually analyzed satellite images to ascertain if a fire had occurred prior to downloading them for fire map creation. If the probable fired occurred before the earliest Planet database records, we instead visually inspected images on Google Earth and Landsat. Since all periods that showed signs of fire were within the last decade, we utilized the Planet database, with its high-resolution imagery and high temporal frequency, to generate the final fire maps. This was achieved through a supervised object-based classification algorithm using the same Simple Linear Iterative Clustering and multilayer perception artificial neural network algorithms for segmentation and feature classification on the differenced NDVI images taken immediately before and after the fire.

## Forest structural data

In this study, we used GEDI level 2B version 2 metrics and selected plant area index (PAI), canopy cover (CC), and canopy height (CH) as variables to examine the impact of fire history on forest structure in the area around Ampijoroa Forest Station and Ambodimanga village of Ankarafantsika National Park. These three variables are important descriptors of the three-dimensional distribution and complexity of vegetation structure that together explain above ground biomass, and in turn important in studies of fire impact on forest structure [35, 56, 57]. Each variable is calculated from the strength of the returned lidar waveform from plant material and are widely accepted data products [35]. PAI quantifies the total plant surface area of above-ground plant matter (leaves, branches, and stems) per unit of ground area [58], which corresponds to the vertical summation of foliage density between, in the case of our data, 5-meter intervals. CC as reported by GEDI corresponds to the percentage of ground coverage by the vertical projection of canopy material [58], which is also be described as "canopy

fractional cover" or the proportion of area that are not canopy gaps [58]. CH is derived from the GEDI "rh100" biophysical attribute, which is the maximum canopy height above ground within a GEDI footprint which is ca. 25-m diameter circle [58]. These biophysical attributes are widely used to monitor stand health and quality and thus are excellent variable to assess fire impacts on the vegetation and its post-fire recovery. The recent availability of high resolution from GEDI makes such analysis much more accessible and provides substantially more information about the forest and post-fire recovery than historically common measures of vegetation health that use spectral indices alone (such as NDVI). After downloading all GEDI footprints that intersected the area of interest, we filtered these footprints to remove low-quality data points and non-forest areas as determined by the 1990 forest-non-forest base map. This resulted in a total of 1,458 footprints from eight different orbits between 2019 and 2023. These footprints were then analyzed for forest structure based on their observation date relative to the fire history at that location.

## Statistical analysis

The fire history maps provided the data to create general descriptive statistics of fire occurrence in the study area and to characterize fire effects on forest structure. We calculated the total number of fires, as well as the total area burnt, the ratio of previously unburnt to newly burnt areas, and the burn intensities of each fire. Burn intensities were generated using the maximum normalized burn ratio (NBR) values calculated from all Landsat data collected during the month of each fire.

To characterize the effects of fire on forest structure, we used a combination of statistical tests (non-parametric options when variance assumptions could not be met) and mixed linear models. Specifically, we conducted non-parametric Kruskal-Wallis tests and Dunn's post-hoc test with Bonferroni correction to examine differences in forest structural attributes across varying burn histories, including the number of fires.

To explore the impact of repeated fire occurrence and time since the last fire on forest structure, we constructed linear mixed effects models [59] with plant area index, canopy cover, and canopy height as dependent variables and the number of fires and the number of months since the last fire as independent variables (fixed effects). Given the heterogeneous management practices, tree species compositions, and climatic conditions (particularly the distinct seasonality in wet and dry periods), we incorporated observation season (dry and rainy seasons) and park management zone (services, community, and buffer zones) as random effects in our models. Data from the peripheral management zone, which is situated outside the protected area of the park and is subjected to a high degree of anthropogenic disturbance, were excluded from the statistical analysis.

## Supporting information

**S1 Table. Kruskal-Wallis (chi-statistic and p-value) and Dunn tests (p-values) on plant area index, canopy cover, and canopy height across varying numbers of forest fires.** (DOCX)

**S2 Table. Linear mixed models testing for the effects of the number of fires (n_fires) and the number of months since the last fire (n_months) on three attributes of forest structure: Plant area index, canopy cover, and canopy height.** Also included in the model are two random variables: the park's management zone (community, buffer, or service) and season (dry season May-October vs. wet season November-April). (DOCX)

## Acknowledgments

We are most grateful to the Director, Mandimby Heriniaina Andriambolona, the Chief of Operation, Aina Dimbilala Rabearivony, and other staff of Ankarafantsika National Park for their outstanding support during our field work. Estoria, Fanamby, Rova, Rado, Alpha, and Jean De were excellent field guides who provided invaluable support. We thank Yusuke Onoda for his advice on the drone survey and other forest canopy analyses, Tomohiro Nishigaki for soil type classification, Ren Kurokawa and Marina Sarindra for logistical support, and Yutaro Fujimoto for his help in acquiring historical aerial imagery.

## Author Contributions

**Conceptualization:** Joseph Emile Honour Percival, Hiroki Sato, Kaoru Kitajima.

**Data curation:** Joseph Emile Honour Percival.

**Formal analysis:** Joseph Emile Honour Percival.

**Funding acquisition:** Hiroki Sato, Kaoru Kitajima.

**Investigation:** Joseph Emile Honour Percival, Hiroki Sato, Tojotanjona Patrick Razanaparany, Ando Harilalao Rakotomamonjy, Zo Lalaina Razafiarison, Kaoru Kitajima.

**Methodology:** Joseph Emile Honour Percival, Hiroki Sato, Kaoru Kitajima.

**Project administration:** Hiroki Sato, Kaoru Kitajima.

**Resources:** Hiroki Sato, Kaoru Kitajima.

**Software:** Joseph Emile Honour Percival.

**Supervision:** Hiroki Sato, Kaoru Kitajima.

**Validation:** Joseph Emile Honour Percival, Hiroki Sato, Tojotanjona Patrick Razanaparany, Kaoru Kitajima.

**Visualization:** Joseph Emile Honour Percival, Hiroki Sato.

**Writing – original draft:** Joseph Emile Honour Percival.

**Writing – review & editing:** Joseph Emile Honour Percival, Hiroki Sato, Tojotanjona Patrick Razanaparany, Ando Harilalao Rakotomamonjy, Zo Lalaina Razafiarison, Kaoru Kitajima.

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
