## [Decision Letter · Decision Letter 0]

17 Oct 2023

PONE-D-23-24765Non fire-adapted dry forest of Northwestern Madagascar: escalating and devastating trends revealed by remote-sensing dataPLOS ONE

Dear Dr. Percival,

Thank you for submitting your manuscript to PLOS ONE. After careful consideration, we feel that it has merit but does not fully meet PLOS ONE’s publication criteria as it currently stands. Therefore, we invite you to submit a revised version of the manuscript that addresses the points raised during the review process.

We look forward to receiving your revised manuscript.

Kind regards,

RunGuo Zang

Academic Editor

PLOS ONE

5. We note that Figure 4 in your submission contain copyrighted images. All PLOS content is published under the Creative Commons Attribution License (CC BY 4.0), which means that the manuscript, images, and Supporting Information files will be freely available online, and any third party is permitted to access, download, copy, distribute, and use these materials in any way, even commercially, with proper attribution. For more information, see our copyright guidelines: http://journals.plos.org/plosone/s/licenses-and-copyright.

a. You may seek permission from the original copyright holder of Figure 4 to publish the content specifically under the CC BY 4.0 license.

6. We note that Figures 1 and 2 in your submission contain [map/satellite] images which may be copyrighted. All PLOS content is published under the Creative Commons Attribution License (CC BY 4.0), which means that the manuscript, images, and Supporting Information files will be freely available online, and any third party is permitted to access, download, copy, distribute, and use these materials in any way, even commercially, with proper attribution. For these reasons, we cannot publish previously copyrighted maps or satellite images created using proprietary data, such as Google software (Google Maps, Street View, and Earth). For more information, see our copyright guidelines: http://journals.plos.org/plosone/s/licenses-and-copyright.

a. You may seek permission from the original copyright holder of Figures 1 and 2 to publish the content specifically under the CC BY 4.0 license. 

Additional Editor Comments:

Please make point by point responses to the comments ot the referees

Reviewers' comments:

Reviewer's Responses to Questions

**Comments to the Author**

1. Is the manuscript technically sound, and do the data support the conclusions?

Reviewer #1: Yes

Reviewer #2: Yes

2. Has the statistical analysis been performed appropriately and rigorously? 

Reviewer #1: Yes

Reviewer #2: Yes

3. Have the authors made all data underlying the findings in their manuscript fully available?

Reviewer #1: Yes

Reviewer #2: Yes

4. Is the manuscript presented in an intelligible fashion and written in standard English?

Reviewer #1: Yes

Reviewer #2: Yes

5. Review Comments to the Author

Reviewer #1: Page 1, Title, which remote-sensing data? Need to be specific?

Line 24, Abstract: how to define “tropical dry forests”? what is the difference between dry and wet forests? I am not sure why there are tropical dry forests Northwestern Madagascar is near the sea, is there any uniform vegetation classifications all over the world?

Line 26, 3052 should be 3,052 ? A comma needs for the third byte, and thereafter.

Line 31, what the resolution of Investigation (GEDI) lidar sensor?

Line 40-41, how to differ natural fires from anthropogenic fires? Does it mean ANP's dry forests exhibit higher vulnerability and resilience to natural fires?

Line 61, If the figures less than ten, better use English words, so 6 months should be six months, and thereafter.

Reviewer #2: The ms. quantifies the historical fire patterns in Ankarafantsika National Park and explores the influence of fire on forest structure. The study's findings hold significance in gaining insights into the fire history and its ecological consequences in the study area. However, the study lacks depth, and specific questions and recommendations are outlined below for further improvement.

1. The manuscript should provide an explanation of what "Plant Area Index" is and provide a detailed description of the calculation method.

2. The Methods section should contain a comprehensive description of "Canopy Cover" and "Canopy Height."

3. The figures in the manuscript require significant improvement in terms of quality.

4. As far as I know, "Plant Area Index," "Canopy Cover," and "Canopy Height" are not typically used as indicators of forest susceptibility.

5. The topographical gradient effect on fire regimes is not analyzed in the manuscript, so the relevant statement in the introduction should be revised (p94-97).

6. The analysis of "Canopy Height" should be justified as it is generally less affected by fire and more related to stand quality.

7. The manuscript should encompass an explanation of historical forest fires.

6. PLOS authors have the option to publish the peer review history of their article (what does this mean?). If published, this will include your full peer review and any attached files.

Reviewer #1: **Yes: **Hua-Feng Wang, Hainan University

Reviewer #2: No

---

## [Author Response · Author response to Decision Letter 0]

17 Dec 2023

We would like to thank the editor and all reviewers for taking the time to carefully read our article and give detailed and constructive comments. We provide the following as our responses to their comments to improve this manuscript. We agree with all comments and suggestions, and made the appropriate changes as detailed below. In addition, since our submission there have been several new GEDI observation orbits in the study. Thus, we have incorporated these additional data in our latest revision to increase the sample size, which resulted in minor changes the observation counts and figures. The major results remain unchanged, and the increased sample size, especially for observations that have experience fire over 4 years ago, strengthen the arguments made in this paper and thus we felt were important to include. 

Detailed Response to the Reviewers

Reviewer #1: 

1. Page 1, Title, which remote-sensing data? Need to be specific?

The title is updated to reflect the primary use of Landsat timeseries and GEDI lidar. The new title is: “Non fire-adapted dry forest of Northwestern Madagascar: escalating and devastating trends revealed by Landsat timeseries and GEDI lidar data”

2. Line 24, Abstract: how to define “tropical dry forests”? what is the difference between dry and wet forests? I am not sure why there are tropical dry forests Northwestern Madagascar is near the sea, is there any uniform vegetation classifications all over the world?

We added a definition to the opening sentence (line 44) that provides a simple and brief explanation of tropical dry forests. This should allow the reader to better contextualize the vegetation and climate of NW Madagascar. We also added forests as a layer to figure 1 so that forest cover can be seen over the rainfall gradient. 

3. Line 26, 3052 should be 3,052 ? A comma needs for the third byte, and thereafter.

We added a comma here and in all appropriate instances thereafter.

4. Line 31, what the resolution of Investigation (GEDI) lidar sensor?

We added the resolution (25-m diameter) to the methodology section (line 344) where GEDI is introduced as a data source.

5. Line 40-41, how to differ natural fires from anthropogenic fires? Does it mean ANP's dry forests exhibit higher vulnerability and resilience to natural fires?

We added more text to the discussion section, see from line 220, that highlights important differences between natural wildfires and those set by people. In general, anthropogenic fires differ from natural fires in the timing and frequency. Anthropogenic fires differ in intensity and frequency than natural fires, and when fire occurs in forest that historically do not burn, it can burn intensely due to high dry fuel load (from dead trees and drier conditions during the dry season). This section supplements the existing discussion on the influence of increased fire frequency in the region. 

6. Line 61, If the figures less than ten, better use English words, so 6 months should be six months, and thereafter.

We went through the manuscript and updated all numerical figures where appropriate.

Reviewer 2

1. The manuscript should provide an explanation of what "Plant Area Index" is and provide a detailed description of the calculation method.

We added several sentences from line 392 to the methods section that describes plant area index and its calculation. We also provide a reference to the Algorithm Theoretical Basis Document that provides the mathematical definitions and a highly detailed calculation methodology.

2. The Methods section should contain a comprehensive description of "Canopy Cover" and "Canopy Height."

Similarly, we also added several sentences to the methodology, here specifically from lines 395, that describe and define canopy cover and canopy height as they are used in this study. These also refer to the same Algorithm Theoretical Basis Document. 

3. The figures in the manuscript require significant improvement in terms of quality.

Each of the figures were completely re-done to improve quality. 

In Figure 1, we removed the Landsat image in sub-panel B for increased readability and interpretability since the purpose of B is to see the area in the context of Madagascar (which is given in A), and is an enlarged portion of an area marked in A. We also added forest cover of 2022 using the methodology of Vieilledent et al. 2018. Finally, the rendering of subplots C and D were improved and plotting artefacts from the original figure were removed. We also added lines to join subplots to show them as enlarged areas from lower subplots. Finally, we replaced the Planet images of C and D with Landsat images to avoid copyright issues.

In Figure 2, we removed plotting artifacts from the first subplot and improved the legend quality. We also improved the overall quality of the other subplots and standardized label size. We also replaced the Planet images of A with a Landsat image to avoid copyright issues.

In Figure 3, we replaced the violin plots with box plots to show the effects of the number of fire as it is easier for potential readers to see statistical differences among groups (A, B, C). We also re-labelled each subplot so that each can be identifiable in the text and caption. All labels and text fonts and font sizes were standardized with other figures and compact letter displays were made more uniform. The counts of observations for each group are different from the initial submission as we were able to include more newly released data from GEDI for the years 2022 and 2023. For subplots D, E, and F, we replaced the violin plots with scatter plots with regression lines for the number of fires separately. As time since fire is a continuous variable, this presentation is statistically more appropriate and consistent with the General Linear Model output shown in Supporting Information S1 Table 2. 

In Figure 4, we replaced the subplot labels so that they match the style and format of other figures.

4. As far as I know, "Plant Area Index," "Canopy Cover," and "Canopy Height" are not typically used as indicators of forest susceptibility.

Thank you for pointing out the typical indicators of forest susceptibility to fire. Our intention was to use them as indicators of forest structure. While plant area index, canopy cover, and canopy height have not traditionally been the primary measures of recovery post-fire, we believe their inclusion offers valuable insights due to advancements in remote sensing technology that now allow for more accurate estimations. To clarify their relevance, we have expanded the discussion in lines 388 to 404 and included additional references that support their use in post-disturbance studies, namely Bolton et al. 2015 and Viana-Soto et al. 2022.

5. The topographical gradient effect on fire regimes is not analyzed in the manuscript, so the relevant statement in the introduction should be revised (p94-97).

This section was revised, and we removed the statement referring specifically to the topographically gradient.

6. The analysis of "Canopy Height" should be justified as it is generally less affected by fire and more related to stand quality.

Thank you for your comment on canopy height. We agree that it is a critical parameter, often indicative of stand quality and less directly affected by fire. To clarify its relevance to our study, we have added a detailed justification in the methodology section (lines 388 and 191). Furthermore, we have expanded our discussion in the results section (line 189) to explain how the data from 2022 and 2023 illuminate the observed decrease in canopy height. When trees die from fire, they remain standing but eventually fall. So, fire erodes the canopy height, an important measure of stand quality. 

7. The manuscript should encompass an explanation of historical forest fires.

We added a brief discussion of natural vs anthropogenic fires to the discussion section from line 220. This should supplement the introduction to historical forest fires in the region that can be found in the paragraph starting at line 85. Although fire is very common across the landscape in western Madagascar, as we point out in our study, it remains uncertain how the impacts of these fires vary across different regions and ecosystems. We are not aware of existing studies that utilize long-term remote sensing time series datasets to examine the effect of fire on forest structure in the dry forests of Madagascar, and few studies exist that look at historical fires in this region. Thus, we feel that this paper offers important insights into the effects of fire in one of the largest remaining dry forests and stands to provide more information on the fire history of the region. 

References:

Vieilledent G, Grinand C, Rakotomalala FA, Ranaivosoa R, Rakotoarijaona J-R, Allnutt TF, et al. Combining global tree cover loss data with historical national forest cover maps to look at six decades of deforestation and forest fragmentation in Madagascar. Biol Conserv. 2018;222: 189–197. doi:10.1016/j.biocon.2018.04.008

Viana-Soto A, García M, Aguado I, Salas J. Assessing post-fire forest structure recovery by combining LiDAR data and Landsat time series in Mediterranean pine forests. Int J Appl Earth Obs Geoinformation. 2022;108: 102754. doi:10.1016/j.jag.2022.102754

Bolton DK, Coops NC, Wulder MA. Characterizing residual structure and forest recovery following high-severity fire in the western boreal of Canada using Landsat time-series and airborne lidar data. Remote Sens Environ. 2015;163: 48–60. doi:10.1016/j.rse.2015.03.004

---

## [Editor Report · Decision Letter 1]

27 Dec 2023

Non fire-adapted dry forest of Northwestern Madagascar: escalating and devastating trends revealed by Landsat timeseries and GEDI lidar data

PONE-D-23-24765R1

Dear Dr. Percival,

We’re pleased to inform you that your manuscript has been judged scientifically suitable for publication and will be formally accepted for publication once it meets all outstanding technical requirements.

Kind regards,

RunGuo Zang

Academic Editor

PLOS ONE

Additional Editor Comments (optional):

accept
---

## [Editor Report · Acceptance letter]

10 Feb 2024

PONE-D-23-24765R1 

PLOS ONE

Dear Dr. Percival, 

I'm pleased to inform you that your manuscript has been deemed suitable for publication in PLOS ONE. Congratulations! Your manuscript is now being handed over to our production team.

Kind regards, 

on behalf of

Professor RunGuo Zang 

Academic Editor

PLOS ONE